# 1st Visual Inductive Priors for Data-Efficient Deep Learning workshop at ECCV 2020: semantic segmentation Challenge Track Technical Report: Multi-level tail pixel cutmix and scale attention for long-tailed scene parsing

Weitao Chen     Zhibing Wang[1]

[1]Alibaba Group
hillskyxm@gmail.com    zhibin.waz@alibaba-inc.com

**Abstract.** CutMix has been proposed to improve the image classification model robustness against input corruptions and its out-of-distribution detection performances. In this work, we propose a new multi-level tail pixel cutmix which will cut regions according to the tail pixel distribution for improving the long-tailed Scene Parsing. After getting new CutMix-ed images, we furthermore apply a scale attention model with multi scale augmentation on these images to get a more effective representation on long-tailed pixel distribution.

## 1 Introduction

Long-tailed label distributions exist in real world commonly. Deep neural networks have been found to perform poorly on less represented classes. For classification tasks, there are two strategies: re-sampling and re-weighting to improve performance on less represented classes. But for a scene parsing task, when we sample pixels of a class in the whole image level, we also get pixels of other classes. So its hard to get a balanced percent of pixel for all classes. And if we use a data augmentations commonly used in scene parsing task such as resize with random crop, the weight of pixels for all classes will be changed. So its not a good solution to directly use re-sampling and re-weighting for classification tasks in a scene parsing task.

CutMix[1] is a data augmentation which improve the image classification model robustness. We propose a multi-level tail pixel cutmix based on the tail pixel distribution rather than direct use of re-sampling and re-weighting to improve the scene parsing model robustness against the long tail distribution poor performances. Then we combine our new cutmix with a scale attention model[2] to get more benefit in a long-tailed scene parsing task.

## 2 Related Works

**CutMix**.The recently proposed CutMix regularizer of Yun et al.[1] combines aspects of MixUp[3] and CutOut[4]. MixUp, Cutout, and CutMix improve super-

vised classification performance, with CutMix outperforming the other two. Geoff French et al. [5] combine cutmix with mean teacher[6] for semi-supervised semantic segmentation and get a state-of-art result on Pascal VOC 2012[7] dataset and cityscapes[8] dataset.

**Scale attention**.Attention mechanism for multi-scale is proposed firstly by Chen et al.[9] to combination multiple scales. Recently, Andrew Tao et al.[2] propose a more effective attention mechanism and get a state-of-art result on cityscape dataset.

## 3    Multi-level tail pixel cutmix

Our multi-level tail pixel cutmix starts from the current label distribution and a baseline model M0.

Firstly we just compute out the label distribution of all scene classes and put them into three buckets: head bucket, middle bucket and tail bucket. Then we train and evaluate a baseline model with the original data and label. If some class in middle bucket perform a poor result, we just put it into tail bucket from the mille bucket. If some class in tail bucket perform a good result, we just put it into middle bucket from the tail bucket. Thus we get the final tail bucket.

Assuming the tail bucket has n class, we can get n tail subsets from the original dataset. Let X and Y denote a training image from original dataset and its label, respectively. We randomly sample one class b from tail bucket and get a subset as B. Let AUGN denote data augmentations such as random crop which may loss the region containing the class b, Let AUGP denote data augmentations such as flip which will keep the region containing the class b. Let AUG denote data augmentations combined with AUGN and AUGP.

For level 0 cutmix, we randomly sample the whole image XB and its label YB from B. Let CYC denote a data augmentations cycle . When the label after cycle contains class b, we stop the cycle. The combining operation as:

$$X_a, Y_a = AUG(X, Y)$$

$$XB_a, YB_a = CYC(AUG(XB, YB))$$

$$X\tilde{} = (1 - M) * X_a + M * XB_a$$

$$Y\tilde{} = (1 - M) * Y_a + M * YB_a$$

$$M = (YB_a == b) * 1$$

For other level cutmix, we first get:

$$MB = (YB == b) * 1$$

Then we get minimum area rectangles R of regions containing class b from MB. R contains x1, y1, w, h. The x1 denote the left x-coordinates, the y1 denote the upper y-coordinates, the w denote the width and the h denote the height of R. For level1 cutmix, we just use x1, y1, w, h to crop all YB and XB from subset

B. The cropped images and labels then are put into a new subset B1.
For level t cutmix, we expand x1, y1, w, h with ratio r and step s:

$$x1 = x1 - (r + s * t) * w$$

$$y1 = y1 - (r + s * t) * h$$

$$w = w + 2 * (r + s * t) * w$$

$$h = h + 2 * (r + s * t) * h$$

$$0 < r < 1, s > 1, and, t > 1$$

We clip new x1, y1, w, h to make sure them are in image and apply them to crop all YB and XB from subset B. Then the cropped images and labels are also put into subset B1.

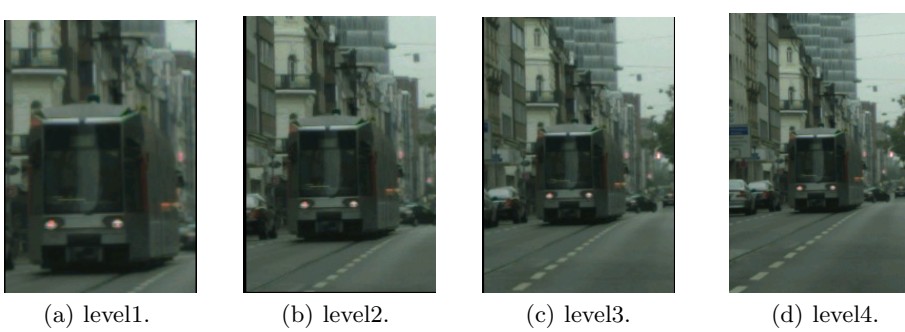

(a) level1.        (b) level2.        (c) level3.        (d) level4.

**Fig. 1.** multi-level pics in subset B1

For this multi-level cutmix, we just randomly sample a image XB1 and its label YB1 from B1 and randomly sample a position P from X. The combining operation as:

$$X_a, Y_a = AUG(X, Y)$$

$$XB1_a, YB1_a = AUGN(XB1, YB1)$$

$$X_a(P) = XB1_a$$

$$Y_a(P) = YB1_a$$

$$X^{\sim} = X_a$$

$$Y^{\sim} = Y_a$$

We use the AUGN to make sure XB1 and YB1 are in the range of $X_a$ and $Y_a$. In each training iteration, multi-level tail pixel cutmix-ed sample $X^{\sim}, Y^{\sim}$ is generated by combining randomly selected two training sample from original dataset and tail subsets.

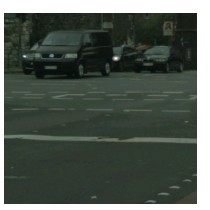 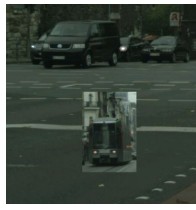 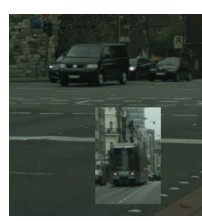 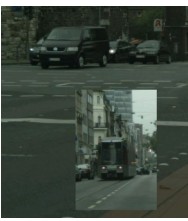

(a) original pic.   (b) level1 cutmix pic.  (c) level2 cutmix pic.  (d) level3 cutmix pic.

**Fig. 2.** multi-level tail pixel cutmix-ed pics

## 4    Scale attention

We use hrnet[10] as our base model. We remove the original last semantic predictions layer and use a dedicated fully convolutional head consisting of $(3*3conv)(BN) \to (ReLU) \to (3*3conv) \to (BN) \to (ReLU) \to (1*1conv)$ as new semantic predictions layer. For scale attention, the same scale attention structure as mentioned in [2] is used in our method. Furthermore we obverse that not all the classes benefit from the whole scales such as train. So for these classes, we just use the scales when we use in training.

## 5    Experiments

We evaluate our method on MiniCity dataset. We expand the original dataset by e times and train a basic model with all tail subsets level 0 cutmix from scratch. Then we get model M1 and use M1 as pre-trained model for training M2-j model with the jth tail subset multi-level cutmix. We use M1 result Mr1 and M2-j result M2rj to get the final result MR.Our ensemble method is a operation similar with cutmix. The operation as:

$$M2j = (M2rj == j) * 1$$

$$MR = Mr1 * (1 - M2j) + M2rj * M2j$$

Our method get a competitive result on MiniCity dataset:

| Method | Iou Class | iIoU Class | IoU Category | iIoU Category | Accuracy |
|---|---|---|---|---|---|
| Baseline | 0.39 | 0.19 | 0.73 | 0.52 | 0.78 |
| multi-level tail cutmix | 0.66 | 0.45 | 0.88 | 0.74 | 0.83 |

**Table 1.** result on MiniCity

## 6   Conclusion

In this work, we propose a milti-level cutmix based on tail pixel distributions and combine it with a scale attention model to get a competitive result on MiniCity dataset.Furthermore we will consider the spatial distribution prior for more stable perfermance.

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
