# OpenReview forum: "Multi-level tail pixel cutmix and scale attention for long-tailed scene parsing"
_thecvf.com/ECCV/2020/Workshop/VIPriors — Submitted to VIPriors_

### Official Review · AnonReviewer2 · 2020-07-21
**Good technical report**

**Confidence:** 5
**Rating:** 3

**Review:**

[Summary] In 2-3 sentences, describe the key ideas, experiments, and their significance.

The authors combine a novel variant of CutMix augmentation, designed for long-tailed datasets, with multi-scale attention. They show their method outperforms a trivial baseline.

[Strengths] What are the strengths of the paper? Clearly explain why these aspects of the paper are valuable.

- Good submission as a technical report to the challenge.

[Weaknesses] What are the weaknesses of the paper? Clearly explain why these aspects of the paper are weak.

- Not a good paper for the workshop paper track: marginal contribution; no hypotheses; no experiments other than final model vs. challenge baseline
- Not related to visual inductive priors
- Incorrect format

[Overall rating] Paper rating: Clear reject

---

### Official Review · AnonReviewer1 · 2020-07-22
**Effective method, but weak scientific contribution**

**Confidence:** 4
**Rating:** 4

**Review:**

#### 1. [Summary] In 2-3 sentences, describe the key ideas, experiments, and their significance.
The paper proposes a multi-scale version of CutMix that boosts the occurance of scarce data classes and combines it with scale attention. Significant performance improvements are shown on the Minicity segmentation dataset.

#### 2. [Strengths] What are the strengths of the paper? Clearly explain why these aspects of the paper are valuable.
* The method seems effective.

#### 3. [Weaknesses] What are the weaknesses of the paper? Clearly explain why these aspects of the paper are weak.
* The technical contribution is marginal and does not have a scientific motivation.
* The explantion of the method is not very clear:
 * How are the classes divided into the buckets based on the label distribution?
 * Why is it necessary to compute the label distribution if the classes are redistributed based on the model performance?
 * How does the scale attention part of the method work? It would have been nice to explain it in the report.
* There are no ablation studies showing the individual effect of multi-scale CutMix and scale attention on the segmentation performance and therefore it is not clear what makes this method effective.

#### 4. [Overall rating] Paper rating
4. OK but not good enough - rejecion

#### 5. [Justification of rating] Please explain how the strengths and weaknesses aforementioned were weighed in for the rating.
Although the proposed method seems effective, the contribution is marginal and is not scientifically motivated.

---

### Decision · Program_Chairs · 2020-07-29

**Decision:**

Reject

**Comment:**

After considering the reviews and further discussion, we do not find sufficient cause to overturn the recommendation of the reviewers.